# FaceDet3D: Facial Expressions with 3D Geometric Detail Hallucination

## Abstract

Facial Expressions induce a variety of high-level details on the 3D face geometry. For example, a smile causes the wrinkling of cheeks or the formation of dimples, while being angry often causes wrinkling of the forehead. Morphable Models (3DMMs) of the human face fail to capture such fine details in their PCA-based representations and consequently cannot generate such details when used to edit expressions. In this work, we introduce FaceDet3D, a method that generates - from a single image - geometric facial details that are consistent with any desired target expression. The facial details are represented as a vertex displacement map and used then by a Neural Renderer to photo-realistically render novel images of any single image in any desired expression and view.

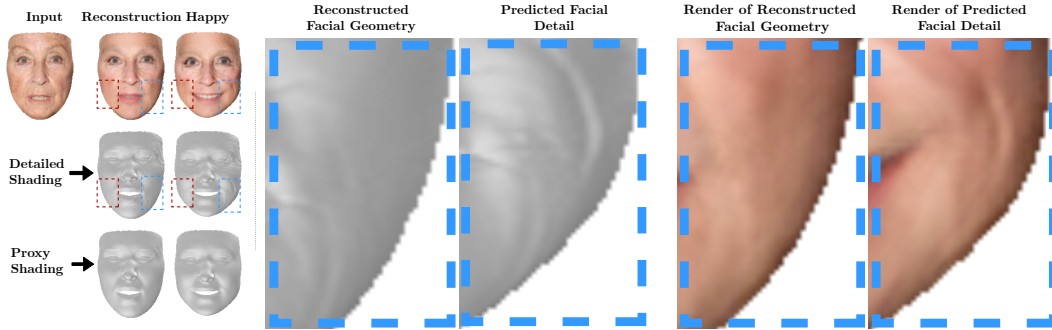

Figure 1: **Facial detail hallucination and rendering.** Given a single input image, a target expression (in this case 'Happy'), and an initial detailed geometry extracted using FDS (Chen et al., 2019) (shown in column 'Reconstruction' and row 'Detailed Shading') as input, our method hallucinates facial geometric details consistent with the target expression. The hallucinated details are added to the smooth proxy geometry (marked as 'Proxy Shading', also extracted using FDS (Chen et al., 2019)), to give a detailed geometry with facial details consistent with the target expression (in column 'Happy' and row 'Detailed Shading'). The detailed geometry is then rendered using Neural Rendering to give the final image (first row of the column labelled 'Happy'). A zoom-in of one of the predicted facial details and its render is shown in column 'Predicted Facial Detail' and 'Render of Predicted Facial Detail' respectively. *(Electronic zoom recommended)*

## 1 Introduction

Modelling the geometry of the human face continues to attract great interest in the computer vision and computer graphics communities. Strong PCA-based priors make 3D morphable models (3DMMs) (Blanz et al., 1999) robust, but at the same time over-regularize them. Thus, they fail to capture fine facial details, such as the wrinkles on the forehead when the eyebrows are raised or bumps on the cheeks when one smiles. Additionally, the lack of diversity in the texture space of most available 3DMMs makes it very hard to generate realistic renderings that capture the large variations of color and texture we observe in human faces. Recent methods (Tewari et al., 2019; 2018; Tran & Liu, 2018; Tran et al., 2019; Zhu et al., 2020; Booth et al., 2017; Dou et al., 2017; Jackson et al., 2017; Kim et al., 2018; Genova et al., 2018) address this by learning richer shape and expression spaces using a variety of data modalities such as in-the-wild images (Tewari et al., 2018; Tran & Liu, 2018; Tran et al., 2019; Zhu et al., 2020) and videos (Tewari et al., 2019). However, despite using more expressive shape and expression spaces, these models still fail to capture fine details in geometry.

Recent methods that accurately estimate facial geometric details from single images (Chen et al., 2019; Tun Trn et al., 2018; Feng et al., 2021), while being unable to hallucinate and photo-realistically render novel details under expression change, have nonetheless enabled the large scale annotation of unpaired image data with facial geometric details. Thus, it is now possible to train facial detail hallucination methods using unsupervised adversarial losses (Choi et al., 2018; Pumarola et al., 2020). Similarly, Neural Rendering (Thies et al., 2019) has made it possible to render 3D geometries with photo-realism via the use of high-dimensional Neural Texture Maps (NTMs). However, NTMs are able to store fine details of the output image, causing the rendered details to be completely independent of the geometric details. They do not change even if the geometric details do, making neural rendering unsuitable for rendering facial geometric details.

In this paper we introduce FaceDet3D, a generative model that hallucinates facial geometric details for any target expression and renders them realistically. The model is made up of two components: 1) A detail hallucination network that infers plausible geometric facial details of a person as their expression changes, from a *single* image of that person. 2) A rendering network that overcomes the aforementioned shortcoming of neural rendering and explicitly conditions the rendered facial details on the *details* of the 3D face geometry. Thus, the rendered details change *only* when the facial geometric details change. This conditioning is achieved through the use of the novel **Augmented Wrinkle Loss** and the **Detailed Shading Loss** during training.

Our method is trained using only a large scale in-the-wild image dataset and a much smaller video dataset captured in controlled conditions, without any 3D data as supervision for the target expression geometry. An exhaustive evaluation shows that once trained, our method is able to generate plausible facial details for any desired anatomically consistent facial expression and render it photo-realistically.

## 2 RELATED WORK

We next describe the most related methods in facial geometry estimation, geometric facial detail estimation and facial expression editing and reanimation.

**Geometric Facial Details Estimation.** Over the past few years there has been significant improvement in the realism of 3D face geometries estimated from single images. In Richardson et al. (2017), a CNN (CoarseNet) first regresses a rough geometry of the face, facial details are then estimated using another CNN (FineNet) using the coarse depth map and input images. In Sela et al. (2017), the regressed correspondence and depth maps are registered onto a template mesh, which is further refined to generate the detailed facial geometry. In Tun Trn et al. (2018) facial details are modelled with bump maps on top of a 3DMM base. Similarly, in Facial Details Synthesis (FDS)(Chen et al., 2019), details are represented as vertex displacements of an underlying 3DMM, trained using a combination of 3D data and in-the-wild images. These methods, however, can only estimate the facial details of the expression manifested in an image, but cannot predict novel facial details for a different expression, which is the motivation of our method. **Geometric Facial Details Animation.** In DECA (Feng et al., 2021), the authors train a network to regress the detailed geometry using a latent detail code. This code enables transfer of details from a target image to the source image. In contrast, the details generated by our method are directly conditioned on expression parameters and action units (Ekman & Friesen, 1978). Further, unlike our method, DECA (Feng et al., 2021) is unable photorealistically render the geometric details, leading to sub-par animation.

**Facial Expression Editing.** The success of image-to-image translation networks (Isola et al., 2017), adversarial training (Goodfellow et al., 2014) and cycle-consistency losses (Zhu et al., 2017) have led to novel expression editing methods that use large-scale in-the-wild training datasets. In (Shu et al., 2017), an unsupervised face model disentangles the input face into albedo, normals and shading. Expressions are then edited by traversals in the disentangled latent space. In Choi et al. (2018), expressions are edited via adversarial losses coupled with cycle consistency. In (Pumarola et al., 2020), a network edits input images to target expressions, represented as Action Units (Ekman & Friesen, 1978). Athar et al. (2020) extends this work by explicitly modeling skin motion followed by texture hallucination. In Choi et al. (2020), template images are used for editing. While these methods give photo-realistic results, they are restricted to the 2D image space and cannot be used to manipulate 3D viewpoint.

# 3 FACEDET3D

## 3.1 PROBLEM FORMULATION

Let $\mathbf{I_x} \in \mathbb{R}^{H \times W \times 3}$ be a face image with an expression $\mathbf{x}$ represented by Action Units (Ekman & Friesen, 1978). Its shape and expression parameters in the Basel Face Model (BFM) (Gerig et al., 2018) space are $\{\boldsymbol{\alpha_s}, \boldsymbol{\alpha_e}\}$. A vertex displacement UV map $\mathcal{D}(\mathbf{I_x}) \in \mathbb{R}^{H_\mathcal{D} \times W_\mathcal{D} \times 3}$ encodes the facial details of the person shown in $\mathbf{I_x}$ with the expression $\mathbf{x}$. We want to: (1) Generate facial details, represented by a vertex displacement map $\mathcal{D}(\mathbf{I_y})$, of the same person with expression $\mathbf{y}$; (2) Render an image of the geometry *with* the generated facial geometric details.

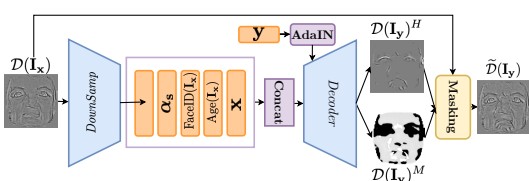

Figure 2: **Detail Hallucination Network.** Given a target expression $\mathbf{y}$, $\mathcal{D}et\mathcal{H}$ outputs a detail hallucination $\mathcal{D}(\mathbf{I_y})^H$ and a detail mask $\mathcal{D}(\mathbf{I_y})^M$. The detail hallucination is combined with the input detail map $\mathcal{D}(\mathbf{I_x})$ using the detail mask to give the final hallucinated facial geometric detail map $\widetilde{\mathcal{D}}(\mathbf{I_y})$.

We use FDS (Chen et al., 2019) to extract the texture map, initial detail map and geometry of the input image $\mathbf{I_x}$:

$$\mathcal{T}(\mathbf{I_x}), \mathcal{D}(\mathbf{I_x}), \boldsymbol{\alpha_s}, \boldsymbol{\alpha_e} \leftarrow \text{FDS}(\mathbf{I_x}) , \qquad (1)$$

where $\mathcal{T}(\mathbf{I_x}) \in \mathbb{R}^{H_\mathcal{T} \times W_\mathcal{T} \times 3}$ is the texture map.

Next, we use the **detail hallucination network**, $\mathcal{D}et\mathcal{H}(\cdot)$, to hallucinate the plausible facial detail map of the person in $\mathbf{I_x}$ for expression $\mathbf{y}$, conditioned on $\mathcal{D}(\mathbf{I_x})$, as follows:

$$\widetilde{\mathcal{D}}(\mathbf{I_y}) = \mathcal{D}et\mathcal{H}(\mathcal{D}(\mathbf{I_x}), \mathbf{x}, \mathbf{y}, \text{Age}(\mathbf{I_x}), \text{FaceID}(\mathbf{I_x})) , \qquad (2)$$

where $\mathbf{y}$ is the target AU, $\mathbf{x}$ is the input AU, $\text{Age}(\mathbf{I_x})$ are features extracted from an age prediction network and $\text{FaceID}(\mathbf{I_x})$ is the facial embedding of $\mathbf{I_x}$ extracted using (Schroff et al., 2015). Note that we do not have access to the ground truth image $\mathbf{I_y}$, thus we hallucinate a plausible detail map of $\mathbf{I_y}$ (i.e $\widetilde{\mathcal{D}}(\mathbf{I_y})$) using $\mathcal{D}et\mathcal{H}$. Once we have $\widetilde{\mathcal{D}}(\mathbf{I_y})$, we use it to displace the vertices along their normal direction giving us the detailed geometry. Finally, we render this detailed face geometry using a **rendering network** $\mathcal{R}(\cdot)$:

$$\tilde{\mathbf{I}}_{\mathbf{y}} = \mathcal{R}\left(\mathcal{T}(\mathbf{I_x}), \widetilde{\mathcal{D}}(\mathbf{I_y}), \boldsymbol{\alpha_s}, \hat{\boldsymbol{\alpha_e}}, \mathbf{y}, \mathbf{c}, l, \boldsymbol{\gamma}\right) , \qquad (3)$$

where $\hat{\boldsymbol{\alpha_e}}$ are the target expression parameters, $\mathbf{c}$ are the desired camera parameters, $\boldsymbol{\gamma}$ is the albedo PCA-space parameters of BFM(Gerig et al., 2018), and $l$ are the lighting parameters.

## 3.2 DETAIL HALLUCINATION

$\mathcal{D}et\mathcal{H}$ takes the input detail map, $\mathcal{D}(\mathbf{I_x})$, the input expression AU $\mathbf{x}$, the shape parameters $\boldsymbol{\alpha_s}$, the face embedding $\text{FaceID}(\mathbf{I_x})$, and the age features $\text{Age}(\mathbf{I_x})$ and extracts features from each of them. There features are concatenated in the channel dimension and passed though the *Decoder* which receives the target action unit $\mathbf{y}$ via Adaptive Instance Normalization (Huang & Belongie, 2017) layers. The *Decoder* gives as output a hallucination $\mathcal{D}(\mathbf{I_y})^H$ and a mask $\mathcal{D}(\mathbf{I_y})^M$ which are combined together with the input detail map, $\mathcal{D}(\mathbf{I_x})$, to give $\widetilde{\mathcal{D}}(\mathbf{I_y})$ as follows:

$$\widetilde{\mathcal{D}}(\mathbf{I_y}) = \mathcal{D}(\mathbf{I_y})^M \odot \mathcal{D}(\mathbf{I_y})^H + (1 - \mathcal{D}(\mathbf{I_y})^M) \odot \mathcal{D}(\mathbf{I_x}) . \qquad (4)$$

The masking mechanism ensures the preservation of the details that are not meant to be changed with the expression. A schematic of the network is shown in Fig 2.

### 3.2.1 TRAINING LOSSES

Due to the lack of a large scale in-the-wild dataset of paired data with expression change or 3D data, we cannot perform regression using ground-truth geometric facial details, $\mathcal{D}(\mathbf{I_y})$ of the image $\mathbf{I_y}$. Therefore, we instead use an adversarial training paradigm along with cycle-consistency losses similar to (Athar et al., 2020; Pumarola et al., 2020) to hallucinate the plausible facial geometric details $\widetilde{\mathcal{D}}(\mathbf{I_y})$ and to ensure the hallucination's fidelity to the target expression and input features. In order to speed up convergence, we weakly supervise the adversarial training using randomly sampled frames from videos of the MUG (Aifanti et al., 2010) and the ADFES datasets (Van Der Schalk et al.,

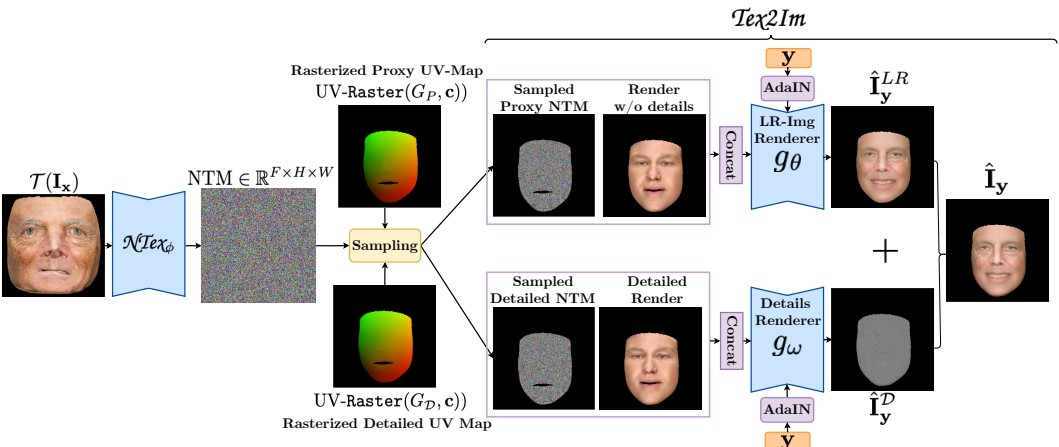

Figure 3: **The Rendering Network.** The Rendering Network, $\mathcal{R}$ first predicts a Neural Texture Map NTM from the given input texture map $\mathcal{T}$ using $\mathcal{N}Tex_\phi$. The NTM is then rasterized using both the proxy (geometry w/o details) and the detailed geometry and input into an image rendering network $Tex2Im$. $Tex2Im$ generates a rendering of the details $\hat{\mathbf{I}}_\mathbf{y}^\mathcal{D}$ and a low-resolution image $\hat{\mathbf{I}}_\mathbf{y}^{LR}$ containing only detail-invariant image texture. They are added together in the final rendered image $\hat{\mathbf{I}}_\mathbf{y}$.

2011), ensuring that the sampling is sparse enough such that there is significant change in expression with frames sampled from each video. We leave the full exposition of all the standard losses and regularizations to the supplementary.

**Expression Adversarial Loss.** In order to ensure the hallucinated facial geometric details, $\widetilde{\mathcal{D}}(\mathbf{I}_\mathbf{y})$, are consistent with the target expression $\mathbf{y}$, as encoded by AUs, we use an expression discriminator $D_{\text{Exp}}$. Given $\mathcal{D}(\mathbf{I}_\mathbf{x})$ of some image $\mathbf{I}_\mathbf{x}$ manifesting expression $\mathbf{x}$, $D_{\text{Exp}}$, outputs the following

$$D_{\text{Exp}}(\mathcal{D}(\mathbf{I}_x)) = \{r, \hat{\mathbf{x}}\} , \tag{5}$$

where $r$ is a realism score and $\hat{\mathbf{x}}$ is the predicted AU. For brevity, we will use $D_{\text{Exp}}(\mathcal{D}(\mathbf{I}_\mathbf{x}))$ and $D_{\text{Exp}}(\mathcal{D})$ interchangeably. We use the Non-Saturating adversarial loss (Goodfellow et al., 2014) along with the R1 gradient penalty (Mescheder et al., 2018) to train $D_{\text{Exp}}$. We use a UNet based discriminator (Schonfeld et al., 2020) in order to discriminate on pixel level. In addition, $D_{\text{Exp}}$ is trained to minimize the error of the predicted AU

$$\mathcal{L}_{\text{AU}}^{D_{\text{Exp}}} = \mathbb{E}_{\mathcal{D} \sim \mathcal{P}_\mathcal{D}} \left[ ||[D_{\text{Exp}}^{\text{AU}}(\mathcal{D}) - \mathbf{x}||_2^2 \right] , \tag{6}$$

where $D_{\text{Exp}}^{\text{AU}}$ is the AU output head of $D_{\text{Exp}}$. The Details Hallucination Network, $\mathcal{D}et\mathcal{H}$, in addition to be trained to minimize adversarial loss, is also trained to minimize the expression loss:

$$\mathcal{L}_{\text{AU}}^{\mathcal{D}et\mathcal{H}} = \mathbb{E}_{\mathbf{I}_\mathbf{x}, \{\mathbf{y}\}} ||D_{\text{Exp}}^{\text{AU}}(\mathcal{D}et\mathcal{H}(\cdot)) - \mathbf{y}||_2^2 \tag{7}$$

where $\mathcal{D}et\mathcal{H}(\cdot)$ is to be read as in Eq. (2) and $\mathbf{y}$ is the target AU.

**Superresolution Losses.** The detail maps generated by FDS (Chen et al., 2019) are of resolution $4096 \times 4096$ and thus cannot be used directly for training due to GPU-memory constraints. To get around this, we train $\mathcal{D}et\mathcal{H}$ on detail maps downsampled to $256 \times 256$. Simultaneously, we finetune a superresolution network, RCAN (Zhang et al., 2018), to super-resolve downsampled $256 \times 256$ patches of $\mathcal{D}(\mathbf{I}_\mathbf{x})$ by a factor of 4

$$\mathcal{L}_{\text{SR}}^{\text{RCAN}} = \text{L1} \left( \text{RCAN} \left( \mathcal{D}(\mathbf{I}_\mathbf{x})_{256}^P \right), \mathcal{D}(\mathbf{I}_\mathbf{x})_{1024}^P \right) , \tag{8}$$

where $\mathcal{D}(\mathbf{I}_\mathbf{x})_{1024}^P$ is a randomly sampled patch of resolution $1024 \times 1024$ from the full-resolution detail map $\mathcal{D}(\mathbf{I}_\mathbf{x})$ and $\mathcal{D}(\mathbf{I}_\mathbf{x})_{256}^P$ is its downsampled version. During inference, we use RCAN twice on the predicted detail map $\widetilde{\mathcal{D}}(\mathbf{I}_\mathbf{y})$ to upsample it to $4096 \times 4096$:

$$\widetilde{\mathcal{D}}(\mathbf{I}_\mathbf{y})^{HR} = \text{RCAN} \left( \text{RCAN} \left( \widetilde{\mathcal{D}}(\mathbf{I}_\mathbf{y}) \right) \right) . \tag{9}$$

In the interest of brevity, we will use $\widetilde{\mathcal{D}}(\mathbf{I}_\mathbf{y})$ in lieu of $\widetilde{\mathcal{D}}(\mathbf{I}_\mathbf{y})^{HR}$ in the remainder of this text.

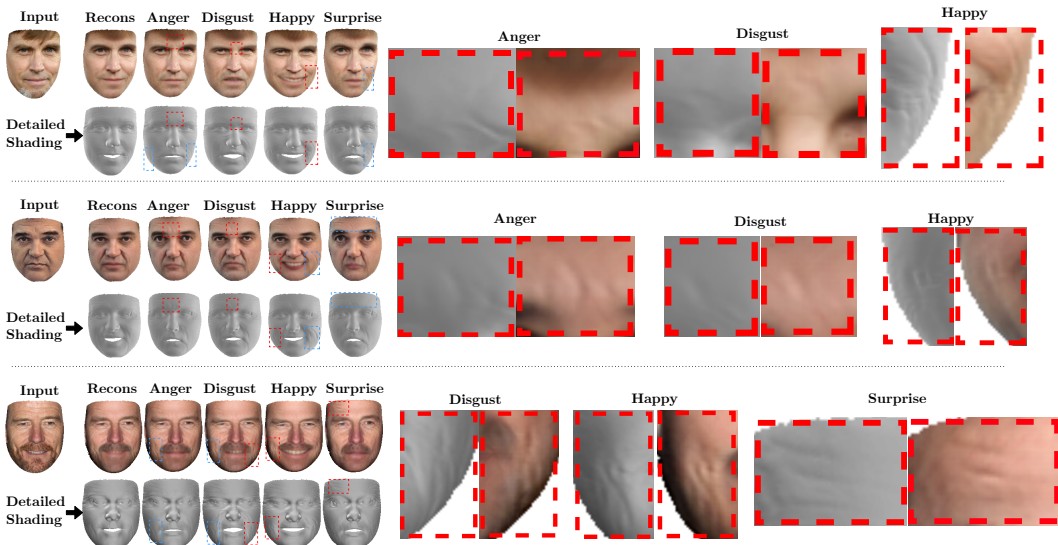

Figure 4: **Expression Change.** Here we show the results of detail hallucination and rendering as the expression changes. The first column is the input image, the second column is the reconstruction (the detailed shading under column 'Recons' is generated by (Chen et al., 2019)) and the subsequent columns are the results of the hallucinated details and their renders by our method under different expressions. The first image row is the output rendering and the second is the shading of the detailed geometry. As one can see $\mathcal{DetH}$ is able to generate realistic details depending on the expression being manifested and $\mathcal{R}$ is able to faithfully render them to the image space. We zoom-in on a subset of details in the final column for greater clarity. *(Please view in high resolution)*

## 3.3 RENDERING NETWORK

The rendering network, $\mathcal{R}$, consists of two subnetworks: (1) The Neural Texture prediction network $\mathcal{NTex}_\phi$ and (2) The image rendering network $\mathcal{Tex2Im}$.

**Neural Texture Prediction.** $\mathcal{NTex}_\phi$ predicts the Neural Textures given the texture map $\mathcal{T}(\mathbf{I_x})$ of image $\mathbf{I_x}$:

$$\text{NTM} = \mathcal{NTex}_\phi(\mathcal{T}(\mathbf{I_x})); \quad \text{NTM} \in \mathbb{R}^{F \times H \times W} , \tag{10}$$

where NTM is the predicted neural texture map with $F$ channels. Unlike in (Thies et al., 2019), where the NTM is identity specific, $\mathcal{NTex}_\phi$ can be used on any $\mathcal{T}(\mathbf{I_x})$, regardless of identitym to generate its corresponding neural texture map. Through training, $\mathcal{NTex}_\phi$ learns to extract the appropriate high-dimensional texture features from $\mathcal{T}(\mathbf{I_x})$ such that NTM can be used to generate a realistic render of the person in $\mathbf{I_x}$ in any desired expression and view.

**Image rendering.** The image rendering network, $\mathcal{Tex2Im}$, consists of two branches, the low-res image renderer $g_\theta$ and the detail renderer $g_\omega$. The low-res image renderer generates identity-specific image textures that are invariant to the predicted geometric details, such as the skin-tone or eye color. The detail renderer $g_\omega$ renders the facial geometric details obtained from the detail hallucination network, $\mathcal{DetH}$, and adds them to the low-res image generated by $g_\theta$. The separation of the image rendering network into two branches allows each branch to focus on its respective task and leads to high-quality renderings.

The low-res image renderer, $g_\theta$, takes as input the NTM sampled using the UV map rasterized by the proxy geometry, $G_P = \{0 \times \widetilde{\mathcal{D}}(\mathbf{I_y}), \boldsymbol{\alpha_s}, \hat{\boldsymbol{\alpha_e}}\}$, i.e the geometry *without* any details, along with the shaded albedo also rasterized by $G_P$:

$$\hat{\mathbf{I}}_{\mathbf{y}}^{LR} = g_\theta \left( \text{Sample}(\text{NTM}, \text{UV-Raster}(G_P, \mathbf{c})), \mathbf{y}, \boldsymbol{\gamma}, l \right) \tag{11}$$

where $\mathbf{c}$ are the camera parameters, $\mathbf{y}$ is the target AU, $\boldsymbol{\gamma}$ is a vector of the coefficients of the albedo PCA-space of the BFM (Gerig et al., 2018), and $l$ are the lighting parameters. Since $g_\theta$ only uses inputs dependent on $G_P$ it generates the image textures that are invariant to details predicted by $\mathcal{DetH}$.

The detail renderer, $g_\omega$, takes as input the NTM sampled using the UV map rasterized by the detailed geometry, $G_\mathcal{D} = \{\widetilde{\mathcal{D}}(\mathbf{I_y}), \boldsymbol{\alpha_s}, \hat{\boldsymbol{\alpha}_e}\}$ along with the shaded albedo also rasterized by $G_\mathcal{D}$:

$$\hat{\mathbf{I}}_\mathbf{y}^\mathcal{D} = g_\theta(\texttt{Sample}(\text{NTM}, \text{UV-}\texttt{Raster}(G_\mathcal{D}, \mathbf{c})), \mathbf{y}, \boldsymbol{\gamma}, l) \tag{12}$$

where $\mathbf{c}$ are the camera parameters, $\mathbf{y}$ is the target AU, $\boldsymbol{\gamma}$ is a vector of the coefficients of the albedo PCA-space of the BFM (Gerig et al., 2018), and $l$ are the lighting parameters. The final output image is calculated as:

$$\hat{\mathbf{I}}_\mathbf{y} = \hat{\mathbf{I}}_\mathbf{y}^{LR} + \hat{\mathbf{I}}_\mathbf{y}^\mathcal{D} . \tag{13}$$

Since all of the detail invariant textures are already generated by $g_\theta$ in $\mathbf{I}_\mathbf{y}^{\hat{LR}}$, the detail renderer, $g_\omega$, can solely focus on realistically rendering the details hallucinated by the details hallucination network $\mathcal{D}et\mathcal{H}$.

### 3.3.1 TRAINING LOSSES

The renderings generated by $\mathcal{R}$ must: 1) faithfully render the geometric details onto the RGB space and 2) be realistic. Neural Rendering (Thies et al., 2019) is designed to address 2) as the high-dimensional neural texture map is able store the fine details of the output texture. Consequently, the details on the rendered image become entirely conditional on the input texture map, $\mathcal{T}(\mathbf{I_x})$, and ignore the detailed geometry $G_\mathcal{D}$. This significantly hurts 1) causing the details on the rendered image to remain unchanged even if the facial geometric details change due to changes in facial expression. In order to ensure the output renderings satisfy 2) we use, along with the branched architecture discussed in Sect 3.3, an **Augmented Wrinkle Loss (AugW)** and the **Detailed Shading Loss (DSL)** to ensure the geometric details are faithfully rendered onto the output image. Additionally, $\mathcal{R}$ is also trained with a photometric loss and an Expression Adversarial Loss in order to maintain photometric and expression consistency. We leave the full exposition of all the standard losses and regularizations to the supplementary.

**Augmented Wrinkle Loss (AugW).** In order to enforce the rendering of geometric details onto the rendered image we add 'fake' wrinkles to an image $\mathbf{I_x}$ and force $\mathcal{R}$ to generate the same. Given the the detailed geometry of $\mathbf{I_x}$, $G_\mathcal{D} = \{\mathcal{D}(\mathbf{I_x}), \boldsymbol{\alpha_s}, \boldsymbol{\alpha_e}\}$, a geometry with 'fake' details $G_\mathcal{D}^* = \{\mathcal{D}(\mathbf{I_z^*}), \boldsymbol{\alpha_s}, \boldsymbol{\alpha_e}\}$ using the geometric details from some random image $\mathbf{I_z^*}$ and the lighting $l$ of $\mathbf{I_x}$, the 'fake' wrinkles are added as follows:

$$\texttt{Shading}(\mathbf{I_x}) = L_{\text{Sph}}(G_\mathcal{D}, l); \texttt{Shading}^*(\mathbf{I_x}) = L_{\text{Sph}}(G_\mathcal{D}^*, l)$$
$$\mathbf{I_x^*} = \texttt{Shading}^*(\mathbf{I_x}) \times \left(\frac{\mathbf{I_x}}{\texttt{Shading}(\mathbf{I_x})}\right) \tag{14}$$

where $L_{\text{Sph}}$ is the spherical harmonic lighting function and $l$ are the coefficients of the first 9 spherical harmonics. The artificially wrinkled image $\mathbf{I_x^*}$ is now re-rendered using $\mathcal{R}$

$$\hat{\mathbf{I}}_\mathbf{x}^* = \mathcal{R}(\mathcal{T}(\mathbf{I_x}), G_\mathcal{D}^*, \mathbf{x}, c, l)$$
$$\mathcal{L}_{\text{AugW}} = \text{LapLoss}(\hat{\mathbf{I}}_\mathbf{x}^*, \mathbf{I_x^*}) . \tag{15}$$

where, LapLoss is the Laplacian Loss (Ling & Okada, 2006). In order to faithfully reconstruct $\mathbf{I_x^*}$, $\mathcal{R}$ is forced to rely on the detailed geometry $G_\mathcal{D}^*$, since the input texture map $\mathcal{T}(\mathbf{I_x})$, and consequently the neural texture map, *contain no* information about the 'fake' wrinkles.

**Detailed Shading Loss (DSL).** In addition to the Augmented Wrinkle Loss, we also try to predict the shading of the detailed facial geometry from the output rendering $\hat{\mathbf{I}}_\mathbf{x}$

$$\hat{\texttt{Shading}}(\hat{\mathbf{I}}_\mathbf{x}) = f_\theta(\hat{\mathbf{I}}_\mathbf{x})$$
$$\mathcal{L}_{\text{DSL}} = \text{LapLoss}(\hat{\texttt{Shading}}(\hat{\mathbf{I}}_\mathbf{x}), \texttt{Shading}^*(\mathbf{I_x})) , \tag{16}$$

where $f_\theta$ is a small convolutional network (CNN) with only two layers and the shading $\texttt{Shading}^*(\mathbf{I_x})$ is calculated as in Eq. (14). We calculate this loss only over the skin region. Since, $f_\theta$ is a small CNN with limited representational capacity, the details must be quite visible on the rendered image $\hat{\mathbf{I}}_\mathbf{x}$ in order for them to be picked up by $f_\theta$ to generate an accurate shading $\hat{\texttt{Shading}}(\hat{\mathbf{I}}_\mathbf{x})$.

**Expression Adversarial Loss.** In order to ensure that the rendered output conforms to the target expression we use an expression adversarial loss. Given a rendered image, $\hat{\mathbf{I}}_\mathbf{x} = \mathcal{R}(\mathcal{T}(\mathbf{I_x}), G_\mathcal{D}, \mathbf{x}, c, l)$,

| Method Name | FID ($\downarrow$) | FaceID ($\downarrow$) |
|---|---|---|
| FaceDet3D | **25.47** | $1.02e^{-3} \pm 4.14e^{-4}$ |
| DECA | 192.21 | $3.04e^{-3} \pm 6.6e^{-4}$ |

Table 1: Quantitative Comparison with DECA based on FID Score and FaceID distance.

manifesting the expression encoded by AU $\mathbf{x}$ an expression discriminator, $D_{Exp}^{RGB}$, outputs

$$D_{Exp}^{RGB}(\hat{\mathbf{I}}_{\mathbf{x}}) = \{r, \hat{\mathbf{x}}\} , \tag{17}$$

where $r$ is a realism score and $\hat{\mathbf{x}}$ is the predicted AU. We use the Non-Saturating adversarial loss (Goodfellow et al., 2014) along with the R1 gradient penalty (Mescheder et al., 2018) to train $D_{Exp}$. In addition, $D_{Exp}^{RGB}$ is trained to minimize the predicted AU error

$$\mathcal{L}_{AU}^{D_{Exp}^{RGB}} = \mathbb{E}_{\mathbf{I}_{\mathbf{x}} \sim \mathcal{P}_{\mathbf{I}}} \left[ ||[D_{Exp}^{RGB,AU}(\mathbf{I}_{\mathbf{x}}) - \mathbf{x}||_2^2 \right] , \tag{18}$$

where $D_{Exp}^{RGB,AU}$ is the AU output head of $D_{Exp}^{RGB}$. The Rendering Network, $\mathcal{R}$, in addition to be trained to minimize adversarial loss, is also trained to minimize the AU loss

$$\mathcal{L}_{AU}^{\mathcal{R}} = \mathbb{E}_{\mathbf{I}_{\mathbf{x}}} ||D_{Exp}^{RGB,AU}(\mathcal{R}(\cdot)) - \mathbf{y}||_2^2 , \tag{19}$$

where $\mathcal{R}(\cdot)$ is to be read as in Eq. (3).

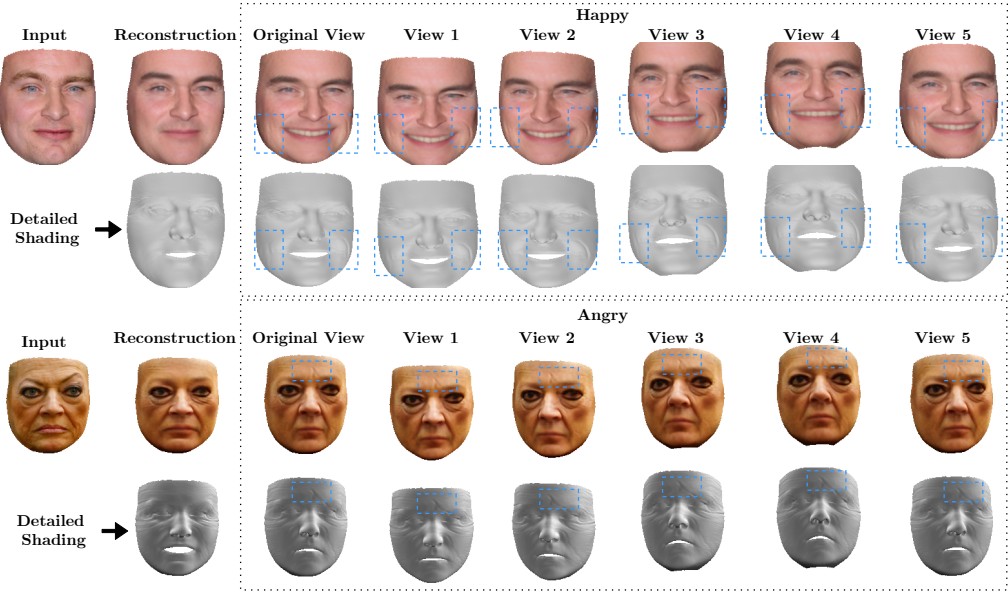

Figure 5: **View Consistency.** In this figure we demonstrate the consistency of the details rendered by $\mathcal{R}$. A subset of the hallucinated details from $\mathcal{DetH}$ are marked with blue rectangles. As can be seen, $\mathcal{R}$ renders the details in a consistent manner across views. *(Please view in high resolution)*

## 4 RESULTS

We train the detail hallucination network, $\mathcal{DetH}$, and the rendering network $\mathcal{R}$, on 9,000 images from the FFHQ dataset (Karras et al., 2019). Additionally, 3,000 frames were sampled from the MUG (Aifanti et al., 2010) and the ADFES (Van Der Schalk et al., 2011) datasets to speed-up training of the detail hallucination network $\mathcal{DetH}$. Due to memory constraints, $\mathcal{DetH}$ and $\mathcal{R}$ are trained independently. Upon publication we will release the code. Below, we show the FaceDet3D's results on expression change, results on view consistency, a comparison with DECA (Feng et al., 2021) and Ablation studies.

**Expression change.** We first demonstrate the detail hallucination and rendering results with expression change. Fig 4 shows the result of changing the expression of some input image to a variety of expressions. The first column of Fig 4 is the input image, the second column shows the image reconstructed using the detail map $\mathcal{D}(\mathbf{I_x})$ predicted by FDS (Chen et al., 2019). In the subsequent columns $\mathcal{D}(\mathbf{I_x})$ is used as input to generate $\widetilde{\mathcal{D}}(\mathbf{I_y})$ where $\mathbf{y} = \{\text{Anger}, \text{Disgust}, \text{Happy}, \text{Surprise}\}$. The first row shows the final rendered image, i.e the output of $\mathcal{R}$ and the second row shows the shaded geometry with the hallucinated details, i.e $G_\mathcal{D}$. The hallucinated details and their corresponding rendering are marked out with dashed red and blue rectangles. We zoom in the red rectangles of the last column. As seen in the second row of Fig 4, the details hallucinated are consistent with the manifested expression. For example, 'Anger' and 'Disgust' (3rd and 4th column of Fig 4 respectively) show consistent wrinkling around the forehead and the nose while 'Happy' (column 6 Fig 4) shows consistent wrinkling around the cheeks. Zooming in the last column of Fig 4 highlights the realism of the hallucinated details produced by the rendering network, $\mathcal{R}$.

Further, in Fig 7 we show the utility of hallucinating details as the expression changes. The first row of Fig 7 shows the image rendered with

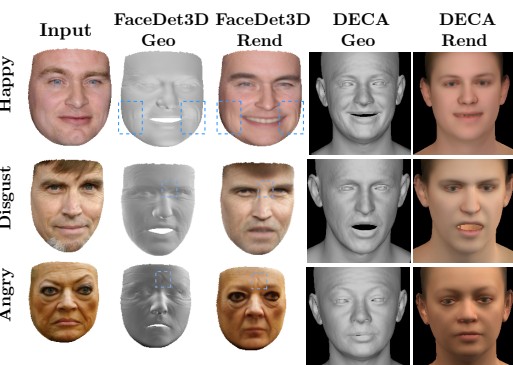

Figure 6: **Qualitative Comparison with DECA** (Feng et al., 2021) The row label shows the expression being animated, the first column is the input image, the second is the detailed geometry hallucinated by FaceDet3D for the expression being animated, the 3rd column is the render generated by FaceDet3D of its hallucinated detailed geometry, the 4th column is the detailed geometry generated by DECA for the expression being animated and the 5th column is render generated by DECA. As can be seen by comparing columns 2 and 4, the details generated by FaceDet3D are significantly more consistent with the expression than those generated by DECA, whose details are more generic (see text for details). Further, a comparison between the 3rd and the fifth column of the figure show that the renders of FaceDet3D are significnatly more realistic than those of DECA. *(Please view in high resolution)*

the hallucinated details from $\mathcal{DetH}$ as the expression changes. The second row of images shows the images rendered without details, this is done by setting the hallucinated detail map to zero. As seen by comparing the skin appearance marked with the red rectangles, in the first row the skin changes realistically as the expression of the person changes due to the changing facial geometric details, while the skin in the second row remains unchanged, significantly hurting realism. In the supplementary we show further results on expression animation and encourage the reader to inspect them.

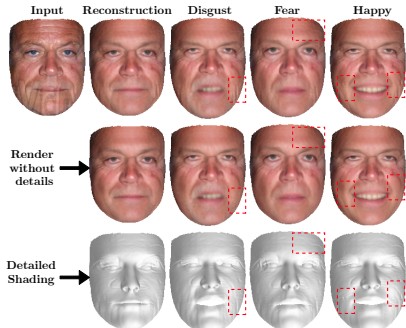

Figure 7: **Details vs. No Details.** We compare the results of changing the expression both with and without hallucinating details. The text in the input image (on the chin) is a watermark. *(Please view in high resolution)*

**View Consistency.** In Fig 5 we show the consistency of the details rendered by $\mathcal{R}$ across various views in novel expressions. While the underlying face model ensures that the detailed geometry is consistent in any view, there is no guarantee its rendering generated by $\mathcal{R}$ would be too. The first column of Fig 5 is the input image, the second column is the reconstruction of the input in the original expression and view and the third column shows the input image manifesting a novel expression in the view of the input image. The subsequent columns show the input image rendered with the target expression in novel views. The blue rectangles around the details in the rendered image show they are rendered with high fidelity to the shaded geometry and therefore look consistent across a variety of views. Fig 5 shows that $\mathcal{R}$ is able to maintain a close to one-to-one correspondence while rendering geometric details to the image space.

**Comparison with DECA** (Feng et al., 2021): In Fig 6 we provide a qualitative comparison of FaceDet3D and DECA. The row label is the expression being animated, the first column is the input image, the second is the detailed geometry hallucinated by FaceDet3D for the expression being animated, the

| Rendering Type | FID ($\downarrow$) | FaceID ($\downarrow$) |
|---|---|---|
| NR Double Branch (Ours) | **25.47** | $1.02e^{-3} \pm 4.14e^{-4}$ |
| NR Single Branch (ablation) | 40.78 | $2.2e^{-3} \pm 3.8e^{-4}$ |
| Ordinary Rendering | 48.18 | $2.7e^{-3} \pm 6.5e^{-4}$ |

Table 2: FID Score and FaceID distance.

3rd column is the render generated by FaceDet3D of its hallucinated detailed geometry, the fourth column is the detailed geometry generated by DECA for the expression being animated and the fifth column is render generated by DECA. Note that DECA's results have the full head because they use the FLAME 3DMM while we use the BFM 3DMM.

By comparing the details in the second and the fourth column of Fig 6 one can see that FaceDet3D generates more concentrated and higher quality details. For example, DECA always generates wrinkles on the forehead regardless of the expression being animated, thus the details generated for 'Happy' (row 1) are inconsistent with the expression. Similarly, a comparison between the third and fifth column of Fig 6 shows that the renders of FaceDet3D are significantly more realistic than those of DECA. In Table 1, we see that the FaceID distance (using FaceNet (Schroff et al., 2015)) and the FID score, across a variety of expression edits, of the renders of FaceDet3D is significantly lower than the renders of DECA. Further, the blue boxes shown in columns 2 and 3 of Fig 6 show that the hallucinated details are faithfully and photo-realistically reproduced in the render.

**Ablation Studies.** We examine the utility of the **Augmented Wrinkle Loss (AugW)** and the **Detailed Shading Loss (DSL)** in rendering the facial geometric details to the image space. Fig 8 shows the results of training $\mathcal{R}$ with and without the **AugW** and **DP** losses. As seen by comparing the results in rows 2 and 3 of Fig 8, without those losses $\mathcal{R}$ fails to render the hallucinated geometric facial details. Next, we ablate using a single network in $\mathcal{T}ex2Im$ ('NR Single Branch') versus using two networks as described in Sect 3.3 ('NR Double Branch') versus using Ordinary Rendering using the provided texture space of BFM(Gerig et al., 2018). We calculate the FID score (Heusel et al., 2017), and the FaceID distance using FaceNet (Schroff et al., 2015) across a variety of expression edits. As can be seen in Table 2, using two networks, one that outputs the detail-invariant textures and the other that renders the details, gives better results as compared to using a single network. It not only produces more realistic images (as measured by the FID score) but also more closely preserves the identity across a variety of expression edits. As it is expected, ordinary rendering performs the worst as its PCA space cannot capture the rich details of the human face across a variety of input identities and expression edits.

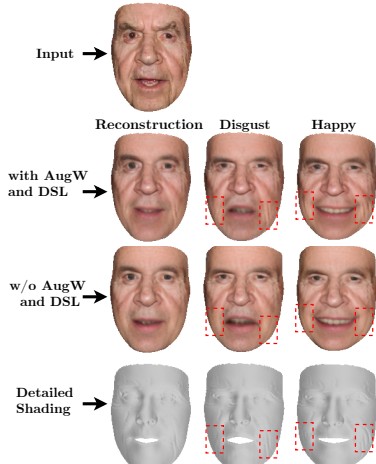

Figure 8: **Ablating AugW and DSL.** We show that without **AugW** and **DSL** details are not rendered to the image space. *(Please view in high resolution)*

## 5 Conclusion and Future Work

We have presented FaceDet3D, a method for hallucinating, from a single image, plausible facial geometric details as the expression changes and render them photo-realistically. The details hallucination network is trained using adversarial losses and weak supervision without any ground-truth 3D data. The rendering network is constrained using the **Augmented Wrinkle Loss** and **Detailed Shading Loss**, forcing it to use cues from the detailed geometry to render the details, ensuring their fidelity and consistency across a variety of expressions and views. The detail hallucination network relies on the detail map estimated by FDS (Chen et al., 2019) as input to predict the plausible details of the target expression, therefore it cannot handle occlusions such as glasses or make-up very well as FDS (Chen et al., 2019) fails in those conditions. In future work, we plan to incorporate explicit disentanglement of lighting in order to enable greater control over the final rendering along with explicit modelling occlusions and joint training of $\mathcal{D}et\mathcal{H}$ and $\mathcal{R}$.

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
