# OpenReview forum: "FaceDet3D: Facial Expressions with 3D Geometric Detail Hallucination"
_ICLR.cc/2022/Conference — ICLR 2022 Submitted_

### Official Review · Reviewer_L964 · 2021-10-31

**Correctness:** 3
**Technical Novelty And Significance:** 3
**Empirical Novelty And Significance:** Not applicable
**Recommendation:** 5
**Confidence:** 3

**Main Review:**

Strengths:
1. This paper proposed a two-stage pipeline that could generate the facial details of any target expressions and render it realistically using just one input image.
2. A detail hallucination network was proposed and trained.
3. The Augmented Wrinkle Loss and Detailed Shading Loss were proposed to train the rendering network.
4. The authors evaluated the method quantitatively and qualitatively, also compared it with the DECA method.
5. The authors further showed that the rendered results are view consistent, proving the effectiveness of the rendering network.
6. Ablation studies are done to show the effectiveness of the two novel losses.
Weaknesses:
1. The readability of this paper can still be improved. For example, a pipeline figure would help in explaining the overall picture of the proposed method; alpha_s is not in the input of Equation (2) although it appears in Fig 2.
2. The figures in this paper could be improved to show the details clearly. For example, in Fig 5, it seems that the lighting/shading of the two subjects' results are quite different, making it hard to see the facial details. The same problem also exists for Fig 4&6. Also,I would suggest the authors to show the zoom-in details in the rectangles as in Fig 1 rather than ask the readers to zoom in for details.
3. The proposed method in this paper is not well-supported by experiments. First, it seems that the image quality of the rendered view is much more blurry than the input image and also the previous work [Chen et al. 2019]. Second, since the proposed method is built on [Chen et al. 2019], it would be fair to include this work for comparison as well reconstructed on the expression of the input image. Lastly, there're no evaluations of the usage of the age prediction network and FaceID embeddings. It would be interesting to know that age information make the proposed method outperforms the others in generating results for people across different ages, etc.

**Summary Of The Paper:**

1. This paper proposes a new method FaceDet3D that can generate geometric facial details from a single image given any desired target expression. The method contains two components: first pass the input image to the Detail Hallucination Network to infer the geometric facial details of the subject as there expression changes; then a rendering network is used to generate the detailed rendering result using the 3D face geometry information.
2. The two component models are trained separately using a large scale in-the-wild images and a small video dataset. During the training of the rendering network, the novel Augmented Wrinkle Loss and Detailed Shading Loss were used.
3. The authors further evaluated the method and compared it to DECA to show that the proposed method could generate plausible facial details for any desired expressions and could render the result photo-realistically.
4. Ablation studies were also done to prove the effectiveness of the components in the proposed method.

**Summary Of The Review:**

This paper proposed a pipeline FaceDet3D that can generate plausible geometric facial details from a single image given any desired target expression. Novel losses were proposed and the model was trained and evaluated to show its effectiveness.
However, I think this paper is still under the bar of acceptance as the experiments are not very convincing. Also, the readability of the paper could still be improved.

---

> ### Author Response · Authors · 2021-11-23
> **Response to Reviewer L964**
>
> Thank you so much for the thoughtful review!
>
> * **Writing:** Thank you so much for the suggestion. We will improve the writing in our paper.
>
>
> * **Figures:** Thank you again for the suggestion! We will revise Figs 4,5,6 to show enlarged details as in Fig 1.
>
>
> * **Comparison with renders of Chen et al.:** We would like to note that Chen et al, do not perform rendering, they use texture mapping from the input image. We provide a comparison with texture mapping [here](https://anonymous.4open.science/r/FaceDet3D-ICLR-4D2C/README.md). As can be seen, the renders generated by texture mapping are inconsistent with the given target expression. They do not contain any geometric details that were generated as a result of expression change. Further, texture mapping from Chen et al. is unable to generate textures such as the teeth. In contrast, FaceDet3D's renders not only contain realistic facial geometric details but are also consistent with the target expression. Further, FaceDet3D is able to generate textures such as teeth.
>
> * **Comparison with details of Chen et al.:** We would like to note that we use the detail map generated by Chen et al as input to DetH to generate the target detail map. In fact, the "Detailed Shading" under the column "Reconstruction" of Figs 1,4,5,7,8 generates the facial details using the detail map generated by Chen et al. This map is then used as input to DetH to generate the facial details shown in subsequent columns for the respective target expressions. Thus, if we were to use the AU of the input detail map as the target AU, the result would be largely unchanged. We will include these results in the paper.

---

### Official Review · Reviewer_6mwj · 2021-11-01

**Correctness:** 3
**Technical Novelty And Significance:** 2
**Empirical Novelty And Significance:** 2
**Recommendation:** 3
**Confidence:** 3

**Main Review:**

Pos:
+ The method produces faces of higher quality when analyses visually (based on the provided examples).
+ While straight-forward, the proposes losses are sound and appear to work well.

Cons:
- The novelty and scope appears limited and incremental. Perhaps the author could better explain&detail this aspect?
- Little to no comparison against current state-of-the-art on 3D face reconstuction.
- The evaluation metrics used are weak in the context of 3D face reconstruction. While the FID score is a good proxy metric for generated images in general it's unclear how representative it is in this scenario. Furthermore the more established point-to-point errors are missing.
- The problem is quite niche, ideally one will want an approach than can in general synthesize high-fidelity models and texture. How can this method be adapted to a more general case?
- What is the reason for using the HQ 4096x4096 data if the images are downsampled to 256x256? Is there a benefit over using 256px directly?
- What is the impact of each proposes loss?

**Summary Of The Paper:**

The paper proposes to improve the quality of the 3D reconstructed faces by taking into account the facial expression. Using a set of generative losses some of the details are recovered or hallucinated. The visual examples provided confirm the improvements, however better numerical evaluations and comparisons are required.

**Summary Of The Review:**

The novelty and scope appear limited and the evaluation is incomplete. The paper could be improved by comparison against state-of-the-art and the use of more standard-metrics.

---

> ### Author Response · Authors · 2021-11-23
> **Response to 6mwj: Clarification regarding the paper**
>
> Thank you for the review. Unfortunately, there seems to be a misunderstanding, our paper does NOT perform facial reconstruction. As noted in the abstract (and in the summaries of Reviewers S1aQ, eEoU, L964), our paper proposes a method that "generates - from a single image - geometric facial details that are consistent with any desired target expression". Our method is an expression based 3D facial detail hallucination method, we would really appreciate it if the reviewer could judge our paper as such.

---

### Official Review · Reviewer_eEoU · 2021-11-03

**Correctness:** 3
**Technical Novelty And Significance:** 3
**Empirical Novelty And Significance:** 1
**Recommendation:** 5
**Confidence:** 4

**Main Review:**

Strengths:
* The motivation is very clear.
* The methodology is well written.
* The use of Age feature and FaceID feature is reasonable and novel
* The idea of separating rendering into two branches is interesting.
* AugW uses the augmented image with wrinkles to enforce the rendering of details.

Weaknesses:
* Lack of ablation study: 1) the “weakly supervise” mentioned in 3.2.1 is not mentioned in the ablation study. 2) The impact of Age and FaceID features is not studied in the ablation. 3) Is the mask necessary in Detail Hallucination Network? 4) the quantitative result from model w/o AugW and model w/o DSL (SEPARATELY) is not provided.

* Based on Figure 5, 6 and 7, I still observe that quite a few details (i.e., wrinkles) that appear in the input are missing in your generated image. It suggests your model still fails to capture the important and significant details when there are substantial wrinkles on the face.

* Performance study should be conditioned on (broken down according to) the key factors, such as age or emotions.  Obviously, the amount of deformation in the face (wrinkles) will increase with age.  Similarly, certain emotions/AUs result in more fine grained deformation than others.

* Comparison with DECA seems to be not fair, as you mentioned, DECA uses different morphable models, the reason may be that input is different.

* There is only one compared baseline, DECA. More compared methods could be added.

* It would be better to separate the contributions of the Detail Hallucination Network and Rendering Network. (e.g. Detail Hallucination Network+existing Rendering method)


**Summary Of The Paper:**

They propose FaceDet3D to generate facial expressions with details. To do this, they propose a Detail Hallucination Network to generate the target detail map from the source detail map along with Expression Adversarial Loss [1] and Superresolution Losses. Then they propose a Rendering Network. In addition to Photometric Loss and Expression Adversarial Loss [1],  they add Augmented Wrinkle Loss and Detailed Shading Loss. The result is an image with target expression containing details (mainly wrinkles), comparison with DECA (a previous work) shows they improve FID and FaceID by large margin.

[1] Conditional Image Synthesis With Auxiliary Classifier GANs


**Summary Of The Review:**

While the authors design two GANs to generate the detail map and final image along with several loss terms, the authors still need to do a thorough and fair ablation study (how important is the “weakly supervise” mentioned in both Section 3.2.1 and Conclusion? How important is AugW and DSL separately?) and compare with fair previous works (How about comparing with models using the same input?).

---

> ### Author Response · Authors · 2021-11-23
> **Response to reviewer eEoU**
>
> Thank you for taking the time to review our paper! We address concerns below
> * **Lack of ablation study:** Thank you, unfortunately due to the lack of time we could not complete the ablation. However, we will include this in the paper.
> * **Details from the input are missing:** Our method is trained on detail maps extracted using Chen et al and also uses them as input tp DetH. Thus, if Chen et al fails to generate some details from the input image, and if those details are expression-independent (such as skin wrinkling etc.), our method is unable to generate them. In Figs 5 and 7, the Detailed Shading under the column name "Reconstruction" is generated using the detail map extracted from Chen et al. and is used as input to generate the detailed shading for the other expressions (the remaining columns of Fig 5 and 7). If this detail map fails to capture certain expression-independent details, our method does not generate them. We believe that accurate extraction of facial-details from in-the-wild images is still an unsolved problem and is a fruitful avenue of future research.
> * **Performance breakdown across key factors:** Thank you so much for this suggestion! We will include this in our paper
> * **Comparison only with DECA**: As far as we know, DECA is the only other published work that hallucinates facial geometric details, conditioned on expression, from a single image. If the reviewer could provide us with references to other work we will be happy to include it. Additionally, we would like to note that FLAME claims to be more expressive than BFM (see Fig 15 and Fig 16 of the [paper]( https://ps.is.tuebingen.mpg.de/uploads_file/attachment/attachment/400/paper.pdf)), thus giving as edge to DECA in terms of the representation they use to learn. However, due to our adversarial paradigm of learning along with the use of generalized neural rendering we are able to generate better quality facial details and render them more realistically.

---

> > ### Comment · Reviewer_eEoU · 2021-11-30
> > **Reply to authors' rebuttal**
> >
> > I appreciate your responses;  I understand the pressure of submitting complete work in time for any submission deadline.  I hope that, once completed, the ablation and the performance context breakdown will offer additional support to your work.

---

### Official Review · Reviewer_S1aQ · 2021-11-03

**Correctness:** 3
**Technical Novelty And Significance:** 2
**Empirical Novelty And Significance:** 2
**Recommendation:** 5
**Confidence:** 4

**Main Review:**

This paper is based on the Basel Face model and much depend upon initial texture and displacement map from Chen et al. The displacement map is trained for the AU but the AUs get changed for normal animation too (like when a person speak in Neutal emotion).Moreover, it also changes for a same person due to the degree of emotion. Because the Network is learning through the adversarial way through the correct generation of AU from the trained displacement map, the network can learn to deform without considering the person specific characteristics and average out the high frequency details. For example, in Fig 4, the last character’s initial face is in happy mood and the deformation on cheek from Chen et al. is quite good in rendering. But that deformation is not so prominent when this method is rendering happy given that initial frame and seems averaged out. But the AUs could be correct. The rendering seems losing the degree of deformation. To show the efficacy of detail displacement map, it would be better to see the result on a face where many wrinkles are present (like an aged person face) like the Fig 1 of Chen et al. Also, if we get a comparison what Chen et al. produce for an expressive face for a video frame (t) and what this method produces given only the AU for that face after initialising with Chen et al. output from an earlier frame (say t-5), then the method’s effectiveness for producing deformation with only AU  can be conceived better.
Regarding the rendering, the concept of producing overall texture followed by addition of detailed texture is almost similar to DECA but the method is different here which utilises the power of neural texture map. As the displacement map is changing the normal of the face and Basel face has the linear albedo subspace, utilization of these two to render a face with many wrinkles(due to emotion) without using NTM can be shown in a figure to qualitatively see the effectiveness of NTM.
The comparison with DECA shows the clear advantage of this method. It would be nice to comment about the utility of FLAME vs Basel face model. FLAME’s PCA component and BASEL’s PCA components are different and hence the identity preservation of the DECA can get deteriorate. Is this improvement due to the method or the choice of the underlying face model? Similarly the expression subspace for FLAME is also different and the impact of that will be in the DECA’s rendering. So it may be better for a fair comparison of the method to state the equivalence or the impact of choice of the underlying 3D face model.


**Summary Of The Paper:**

This paper proposes a pipeline to induce geometrical deformations due to expressions on a person’s 3D face. The existing works either uses single image based 3D reconstruction that manifest the expression present in image or learn a neural representation of the deformation (latent code) which can be used to transfer the deformation due to expression from one face to other. This paper uses direct expression parameters and action units instead of a latent code and predict the deformation on 3D face for an expression instead of manifesting it from image (unlike reconstruction methods).
To this end, the authors use an existing robust single image based 3D face deformation method (Chen et al.) to initialise the texture map and geometry as a displacement map of a 3D face (using Basel Face Model) from an image.  Because there is no ground truth of the new expression, they train it using adversarial loss. They produce the deformation using the Action Units(AU) as input and tries to produce those AU from the output displacement map.
For rendering, the method uses the Neural Texture which is trained with the texture map produced by the Chen et al. The rendering has two components. One is the coarse level rendering which utilises the shape and expression parameters of the Basel Face model and the detail rendering consisting of the information of the displacement map due to the AU obtained through this method. Both are combined to get the final appearance.
They compared the result with DECA (Siggraph 21) which is a FLAME 3D model based method and shows that the identity, shape and expression rendering is better in the proposed method.


**Summary Of The Review:**

The pipeline of the paper can work, but the impact of the method and the novelty is not very clear as it depends heavily on Chen et al. and to some extent like DECA’s principle in rendering. As mentioned in the main review, the dependency of the Chen et al. needs to be critically evaluated as an initialization. Also, the choice of underlying 3D face model and impact of it in the outcome is also needed to understand the method’s utility in the comparisons. In the rendering, it seems the high frequency is getting lost. A few more results with more wrinkles in face would be better to judge the effectiveness of the rendering given a good initialisation from Chen et al.

---

> ### Author Response · Authors · 2021-11-23
> **Response to S1aQ**
>
> Thank you so much for the thoughtful review! Below we address specific critiques:
> * **Learning person specific high frequency details:** We would like to note that our method uses the detail maps generated by Chen et al. for training, thus we are limited by the resolution of details generated by Chen et al. In fig 4, we show the hallucinated details for 3 different identities across 4 expression edits. If one looks at the expression 'Happy', we see that the details on each person are different. Thus our method does indeed generate person-specific details. Further, for each expression, our method generates details not present in the input, as can be seen in the insets of Fig 4.
> * **Comparison on a video:** Thank you so much for the suggestion! We will add this in the paper.
> * **Comparison to BFM texture space:** We provide a comparison to the BFM texture space rendering [here](https://anonymous.4open.science/r/FaceDet3D-ICLR-4D2C/README.md). As can be seen, FaceDet3D's rendering is significantly better.
> * **FLAME vs. BFM:** We do not believe that the difference in results of DECA is due to them using FLAME instead of BFM. .  FLAME claims to produce a better expression space than BFM (see Fig 15 and Fig 16 of the [paper](https://ps.is.tuebingen.mpg.de/uploads_file/attachment/attachment/400/paper.pdf)), thus, in principle, giving an advantage to DECA. However, we use generalized neural rendering to generate our results while DECA only uses the albedo space of BFM, which is why our renders are more realistic.  We would like to note that our method is independent of the underlying 3DMM, we used BFM simply because Chen et al, the erstwhile SOTA, does so.

---

### Decision · Program_Chairs · 2022-01-20

**Decision:**

Reject

**Comment:**

The reviewers raised a number of major concerns including a poor readability of the presented materials, incremental novelty of the presented and, most importantly, insufficient and unconvincing ablation and experimental evaluation studies presented. The authors’ rebuttal failed to address all reviewers’ questions and failed to alleviate reviewers’ concerns. The authors explain that due to the lack of time they could not complete all experimental studies. A major revision of the paper is needed before the paper can be accepted for publication. Hence, I cannot suggest this paper for presentation at ICLR.